# ReMoBA: Representative Replay and Mixture of BatchNoise Autoencoders for Pre-Trained Model-Based Federated Domain-Incremental Learning

## Abstract

Federated Domain-Incremental Learning (FDIL) orchestrates model updates across multiple clients with data drawn from diverse domains. Although pre-trained models (PTMs) offer a robust initial foundation, naively adapting them in FDIL environments often leads to inter- and intra-client task confusion, in addition to catastrophic forgetting. In this work, we mathematically characterize these issues within the FDIL framework and introduce Representative Replay and Mixture of BatchNoise Autoencoders (ReMoBA), a replay-based generative approach that consolidates both representations and classifiers. Specifically, ReMoBA employs a diversity-guaranteed exemplar-selection strategy in the latent space to replay the optimally curated tiny subset of past data stored at the client side, preserving previously acquired representations while the new domain embeddings are determined for all the clients by the server globally via charged particle system energy minimization equations and repulsive force algorithm. ReMoBA further leverages a mixture of autoencoders, trained with structured noise, to enhance robustness and generalization. Extensive experiments on benchmark datasets demonstrate that ReMoBA consistently outperforms state-of-the-art FDIL methods, offering PTMs superior adaptability to new domains and mitigating inter- and intra-client task confusion. Source code will be released upon acceptance.

## 1 Introduction

Federated Domain-Incremental Learning (FDIL) is a challenging scenario in which a model must continuously adapt to new domains on decentralized clients (Sun et al., 2024a; Li et al., 2024a; Sun et al., 2024b; Zhou et al., 2024b). Pre-trained models (PTMs) offer a powerful initialization for such incremental learning tasks (Han et al., 2021; Wang et al., 2022a;b;c). However, simply fine-tuning or naively adapting PTMs in FDIL can cause inter- and intra-client task confusion in addition to catastrophic forgetting, as highlighted in recent studies (Khademi Nori & Kim, 2025; Nori et al., 2025; Shokrolahi & Kim, 2025; Cormerais et al., 2021; Masana et al., 2020).

PTMs in FDIL specifically encounter three major issues: inter-client task confusion, intra-client task confusion, and within-task class confusion (Nori et al., 2025). Inter-client task confusion arises when the model struggles to differentiate tasks originating from distinct domains across various clients. In contrast, intra-client task confusion occurs when the model fails to discriminate among multiple tasks within the same client. Lastly, within-task class confusion occurs when the model fails to discriminate classes within a task. Catastrophic forgetting refers to the tendency of the model to lose either of the three aforementioned *distinctory knowledge* (Khademi Nori & Kim, 2025; Shokrolahi & Kim, 2025; Cormerais et al., 2021; Masana et al., 2020). Addressing these challenges (across the representations and classifiers) is vital to enhancing the effectiveness of PTMs in FDIL environments.

Previous works utilizing PTMs in FDIL scenarios (Sun et al., 2024a; Li et al., 2024a; Sun et al., 2024b; Zhou et al., 2024b) have not adequately addressed critical issues of inter- and intra-client task confusion (Khademi Nori & Kim, 2025; Shokrolahi & Kim, 2025; Cormerais et al., 2021; Masana et al., 2020) (across representations and classifiers). To bridge this gap, our contributions in this paper are as follows:

- We mathematically formalize inter-, intra-client task confusion, and within-task class confusion for PTMs in FDIL, providing theoretical insight into the nature of optimal solutions for FDIL scenarios.

- We propose Representative Replay and Mixture of BatchNoise Autoencoders (ReMoBA), a novel approach that guarantees sample diversity by leveraging a latent-space exemplar-selection strategy to replay a tiny curated subset of past data stored on the client side, effectively addressing intra-client task confusion and within-task class confusion (Bardenet et al., 2024).

- On the server side, ReMoBA utilizes energy minimization equations from charged particle systems and the repulsive force algorithm (Nazmitdinov et al., 2017) to globally determine new domain embeddings for all clients, ensuring consistency of representation among clients.

- ReMoBA further employs a mixture of autoencoders trained with structured noise. These autoencoders have been shown to effectively reduce inter-client task confusion (Nori et al., 2023; van de Ven et al., 2021).

## 2 LITERATURE REVIEW

**Domain-Incremental Learning (DIL).** Incremental learning is divided into three categories: task-incremental learning, DIL, and class-incremental learning (van de Ven & Tolias, 2019). In DIL, the model encounters a sequence of tasks where the input distributions change (different domains) while the output classes remain the same (Wang et al., 2024). A canonical example is learning to recognize the same objects under different imaging conditions or backgrounds, each condition providing data from a new domain (Shi & Wang, 2023).

**DIL with Pre-Trained Models (PTMs).** Leveraging PTMs has emerged as a powerful trend in incremental learning, particularly in DIL scenarios (Wang et al., 2022a;b). PTMs such as large convolutional backbones or vision transformers trained on massive datasets offer broad, generalizable feature representations (Zhou et al., 2024b; Smith et al., 2023). In fact, recent work has shown that using a frozen pre-trained encoder and only learning lightweight components can beat state-of-the-art incremental learning methods (Wang et al., 2022c; Han et al., 2021). For example, a simple baseline that keeps a pre-trained CNN frozen and incrementally computes class prototypes (means of feature vectors for each new class) can outperform complex incremental learning models trained from scratch (Zhou et al., 2024a).

**Federated Domain-Incremental Learning (FDIL).** FDIL integrates DIL (Shi & Wang, 2023) into the federated learning framework, enabling multiple clients (such as devices or data silos) to collaboratively train a shared global model while preserving data privacy (Li et al., 2024a). In FDIL, each client encounters a sequence of tasks or domains over time, with their data remaining local to avoid sharing with others or the central server (Li et al., 2024b). Notably, the task sequences across clients may exhibit distinct domain shifts, reflecting the diverse data distributions typical in decentralized environments (Psaltis et al., 2023). This setup is particularly relevant to real-world applications; for instance, in personal mobile devices, where each device continuously encounters new domains, FDIL allows for collective model improvement without the need to pool sensitive data centrally (Huang et al., 2022).

**Task Confusion.** In incremental learning, models often suffer from task confusion, a challenge where they must discriminate multiple tasks without explicit task identifiers (Cormerais et al., 2021; Masana et al., 2020). Khademi Nori & Kim (2025) demonstrate that task confusion is distinct from catastrophic forgetting. Building upon that, Nori et al. (2025) extend the concept of task confusion to federated learning, distinguishing between inter-client and intra-client task confusion. Recent work by Shokrolahi & Kim (2025) demonstrates that metric learning can effectively mitigate task confusion.

**Exemplar Selection.** Exemplar selection is a key method in incremental learning, designed to retain a small, representative subset of data from prior tasks to address task confusion and catastrophic forgetting (Nokhwal & Kumar, 2023; Luo et al., 2024). Methods vary in complexity, ranging from simple random sampling - surprisingly effective despite its simplicity - to more sophisticated techniques such as clustering or herding, which select exemplars based on proximity to class centroids

(Masana et al., 2020; Pereira et al., 2025). Advanced approaches, however, explicitly emphasize diversity. For example, Determinant Point Processes Probabilistically Select a Guarantee-ably Diverse exemplar set (Chen et al., 2023; Bardenet et al., 2024).

**Generative Modeling.** Incremental learning strategies can be divided into two broad categories: discriminative and generative. Most strategies including generative replay fall within the discriminative category because they ultimately rely on a discriminator for classification—even when generators are employed for rehearsal purposes (Khademi Nori & Kim, 2025). In contrast, generative classifiers employ generative modeling directly for classification. Hayes & Kanan (2020); Nori et al. (2023); van de Ven et al. (2021); Zając et al. (2023) propose a rehearsal-free generative classifier that achieves state-of-the-art performance in addressing task confusion and catastrophic forgetting.

## 3 FDIL PROBLEM FORMULATION

In FDIL, we consider a system consisting of $K$ clients, each progressively encountering sequences of tasks drawn from distinct domains. For client $k$, this sequence is represented as $\{\mathcal{D}_k^t\}_{t=1}^{B_k}$, where $\mathcal{D}_k^t = (\mathcal{X}_k^t, \mathcal{Y}_k^t)$ denotes the dataset for the $t$-th task. Here, $\mathcal{X}_k^t = \{\boldsymbol{x}_{k,i}^t\}_{i=1}^{n_{k,t}}$ consists of input samples, and $\mathcal{Y}_k^t = \{y_{k,i}^t\}_{i=1}^{n_{k,t}}$ are the corresponding labels with each $\boldsymbol{x}_{k,i}^t \in \mathbb{R}^D$ belonging to class $y_{k,i}^t \in \mathcal{Y}$.

In FDIL, the input distribution varies both temporally within a client and spatially across clients. Specifically, for client $k$, the data distribution changes across tasks, i.e., $p(\mathcal{X}_k^t) \neq p(\mathcal{X}_k^{t'})$ for $t \neq t'$, reflecting intra-client domain shifts. Additionally, at the same global task $t$, data distributions differ across clients, i.e., $p(\mathcal{X}_k^t) \neq p(\mathcal{X}_{k'}^t)$ for $k \neq k'$, capturing inter-client heterogeneity. The target of FDIL is to train a global model $f$ that accurately classifies instances from all domains encountered by all clients up to the current task $t$, while the objective function minimizes the expected loss over the union of all seen data:

$$f^* = \underset{f \in \mathcal{H}}{\arg\min} \, \mathbb{E}_{(\boldsymbol{x},y) \sim \mathcal{D}_{\text{union}}^t} \mathbb{I}(y \neq f(\boldsymbol{x})) \quad \text{and} \quad f^* \approx \underset{f \in \mathcal{H}}{\arg\min} \, \mathbb{E}_{(\boldsymbol{x},y) \sim \mathcal{D}_{\text{union}}^t} \mathcal{L}(f(\boldsymbol{x}), y) \quad (1)$$

where $\mathcal{D}_{\text{union}}^t = \bigcup_{k=1}^K \bigcup_{s=1}^t \mathcal{D}_k^s$ represents the union of all seen datasets, $\mathcal{H}$ is the hypothesis space, $\mathbb{I}(\cdot)$ is the indicator function (1 if true, 0 otherwise), and $\mathcal{L}$ is a loss function measuring the discrepancy between the model's predictions and true labels. Since data remains local in FDIL for privacy, this union is conceptual; the server has to achieve this goal through aggregated updates without directly accessing client data.

Following prior work (Wang et al., 2022b;c;a; Smith et al., 2023; Zhou et al., 2024b), we initialize the global model with a pre-trained Vision Transformer (ViT) PTM to leverage its strong feature extraction capabilities across evolving domains while maintaining a shared feature space, which is the last `[CLS]` token. As in (Zhou et al., 2024b), the PTM consists of a embedding function $\phi(\cdot) : \mathbb{R}^D \to \mathbb{R}^d$ that extracts compact feature representations and a classifier $W$, forming the full predictive function $f(\boldsymbol{x}) = W^\top \phi(\boldsymbol{x})$. The classifier is represented by a weight matrix $W \in \mathbb{R}^{d \times tK|Y|}$, where $t$ denotes the current global task index, and $K$ is the number of participating clients in FDIL.

---

**Algorithm 1** Federated Domain Incremental Learning

**Require:** Number of clients $K$, number of global tasks $T$, rounds per task $R$, the PTM $f_0$
1: **for** each global task $t = 1, \ldots, T$ **do**
2:    Server broadcasts model $f_{t-1}$ to clients
3:    **for** each communication round $r = 1, \ldots, R$ **do**
4:       **for** each client $k \in \{1, \ldots, K\}$ **in parallel do**
5:          Receive domain-specific data $\mathcal{D}_k^t$
6:          Perform local training:
7:          $f_{t,k}^r \leftarrow \text{LOCALUPDATE}(f_{t-1}, \mathcal{D}_k^t)$
8:          Send updated local model $f_{t,k}^r$ to server
9:       **end for**
10:    Server aggregates client models:
11:       $f_t \leftarrow \frac{1}{K} \sum_{k=1}^K f_{t,k}^R$
12:    **end for**
13: **end for**

---

The classifier at each client dynamically expands at each task by appending $|Y|$ new class-specific weight vectors, yielding $W = [\mathbf{w}_1, \mathbf{w}_2, \ldots, \mathbf{w}_{tK|Y|}]$. Predictions are made by selecting the class index that maximizes the inner product between the weight vectors and the feature representation $\hat{y} = \left(\arg\max_i \mathbf{w}_i^\top \phi(\boldsymbol{x})\right) \mod |Y|$ (Wang et al., 2022b;c;a; Smith et al., 2023; Zhou et al., 2024b).

As shown in Algorithm 1, the learning process in FDIL is organized into a sequence of global tasks $t = 1$ to $T$, where $T = B$ if all clients have the same number of tasks (a simplifying assumption for

this exposition). At each global task $t$, client $k$ receives data $\mathcal{D}_k^t$ from its $t$-th domain and performs local training. At each communication round $r$, a central server aggregates these local updates to refine a global model $f_t$. Each client accesses only its current dataset $\mathcal{D}_k^t$ during the $t$-th training stage, retaining no direct access to past data unless explicitly curated.

Various approaches have been proposed in the literature to address incremental knowledge incorporation in FDIL. Some methods freeze the PTM feature extractor $\phi(\cdot)$ to preserve the stability of learned representations, which are formed by training on a vast corpus of data. In contrast, other works allow fine-tuning the PTM backbone, reporting promising results in adapting to new tasks while mitigating catastrophic forgetting. In the next section, we present our proposed approach ReMoBA that follows the latter approach, making tiny deliberate adjustments to the learning representations via energy minimization equations from charged particle systems and the repulsive force algorithm to accommodate new domains. Furthermore, to combat intra-client task confusion and catastrophic forgetting, ReMoBA curates a tiny number of samples with guaranteed diversity. And finally, ReMoBA adopts generative modeling to address inter-client task confusion.

## 4 PROPOSED APPROACH: REMOBA

In traditional supervised learning, models are trained on a fixed dataset with a known class distribution, and full access to training data is assumed throughout the training process. The objective is typically to minimize a loss function that measures the discrepancy between predicted and true labels:

$$I_{\boldsymbol{\theta}} = \int_{\mathcal{X} \times \mathcal{Y}} \ell(f_{\boldsymbol{\theta}}(\boldsymbol{x}), y) p(\boldsymbol{x}, y) \, d\boldsymbol{x} \, dy. \tag{2}$$

In contrast, FDIL requires models to evolve continuously across tasks and clients, where data remain decentralized and non-stationary. The global loss function must therefore account for both past and present contributions across all client interactions (Nori et al., 2025). This dynamic is captured in the following recursive formulation:

$$I_{\boldsymbol{\theta}}^{(k+1)} = \sum_{m=1}^{M} \sum_{n=1}^{M} \left( I_{\boldsymbol{\theta}}^{(k),m,n} + \Delta I_{\boldsymbol{\theta}}^{(k+1),m,n} \right), \tag{3}$$

in which the incremental update $\Delta I_{\boldsymbol{\theta}}^{(k+1),m,n}$ is defined as:

$$\Delta I_{\boldsymbol{\theta}}^{(k+1),m,n} = \frac{1}{N^2} \left( \sum_{i=1}^{N} \sum_{j=1}^{N} u_{i_{k+1},j_{k+1}}^{m,n} + 2 \sum_{l=1}^{k} \sum_{i=1}^{N} \sum_{j=1}^{N} u_{i_l,j_{k+1}}^{m,n} \right), \tag{4}$$

where $u_{i_l,j_{k+1}}^{m,n}$ represents the pairwise loss between class $i$ from client $m$ (at task $l$) and class $j$ from client $n$ (at task step $(k+1)$).

Equations 3 and 4 capture the fundamental tension in FDIL: balancing the retention of previously acquired knowledge, denoted by $I_{\boldsymbol{\theta}}^{(k),m,n}$, against the acquisition of new information. This knowledge retention is essential for addressing the stability-plasticity dilemma—the challenge of learning new tasks without catastrophically forgetting previous ones. Common approaches to achieve this balance include regularization techniques that constrain parameter updates, knowledge distillation methods that preserve learned representations, and replay strategies that revisit past examples during new learning phases.

Equation 4 contains two loss components that address different aspects of the learning challenge. The first term governs learning the new task, capturing all interactions between classes in the current task. The second term quantifies the loss between the new task and all previously learned tasks. Inadequate minimization of this second term leads to *intra-client task confusion*, where the model cannot effectively distinguish between different tasks within the same client (Nori et al., 2025).

Beyond this intra-client challenge, Equation 3 implicitly addresses what we term *inter-client task confusion*—the loss arising from interactions between tasks across different clients. When inter-client task confusion remains unresolved, the model loses its ability to differentiate between classes belonging to different clients (Nori et al., 2025).

The above equations therefore reveal four fundamental challenges in federated continual learning: (i) preserving previously acquired knowledge, (ii) effectively assimilating new tasks, (iii) mitigating intra-client task confusion, and (iv) resolving inter-client task confusion. The literature collectively refers to challenges (i) and (iii) as local forgetting, while challenge (iv) constitutes global forgetting (Dong et al., 2022).

We propose ReMoBA–Representative Replay and Mixture of Batch-Noise Autoencoders. The proposed approach systematically addresses all the aforementioned challenges. ReMoBA comprises three pillars:

The first pillar—intended to address old knowledge preservation and intra-client task confusion—is a provably diverse coreset that clients replay locally. We cast exemplar selection as a negative-dependence sampling problem: selecting a minimal subset that is *simultaneously* representative of past domains and maximally non-redundant within the latent space of the ViT backbone. Determinantal Point Processes (DPPs) offer an ideal probabilistic framework by inherently favoring well-distributed points with rigorous variance guarantees. Formally, we adopt the $\varepsilon$-multiplicative coreset definition of Bardenet et al. (2024) and implement the corresponding sampler.

---

**Algorithm 2** DPP-Coreset Sampler for FDIL

**Require:** Data points $X = \{x_1, \dots, x_n\}$; weights $\mu$; ViT feature extractor $\phi(\cdot)$; target size $m$
**Ensure:** Weighted coreset $(S, \omega)$ with $|S| = m$
    **Step 1: Feature extraction and kernel construction**
1: $F \leftarrow [\phi(x_1), \dots, \phi(x_n)]^\top \in \mathbb{R}^{n \times d}$      ▷ ViT features
2: $K_{ij} \leftarrow \dfrac{F_i^\top F_j}{\|F_i\|_2 \|F_j\|_2}$      ▷ cosine similarity kernel
3: Find scaling factor $\alpha$ via bisection s.t. $\mathrm{tr}(\alpha K) = m$; set $K \leftarrow \alpha K$
    **Step 2: $k$-DPP sampling**
4: **if** $n \leq 5000$ **then**
5:     Compute eigendecomposition $K = \sum_{j=1}^n \lambda_j \, u_j u_j^\top$
6:     Sample $S \subset X$ with $|S| = m$ via exact DPP sampling
7: **else**
8:     Use Nyström approximation for scalable sampling to obtain $S$
9: **end if**
    **Step 3: Importance reweighting**
10: **for all** $x_i \in S$ **do**
11:     $\omega_i \leftarrow \mu(x_i)/K_{ii}$
12: **end for**
13: **return** $(S, \omega)$

---

Specifically, given a weighted dataset $(X, \mu)$ and a function class $\mathcal{F} \subseteq \{f : X \to \mathbb{R}\}$, an $\varepsilon$-multiplicative coreset defined as a weighted subset $(S, \omega)$ satisfying, for all $f \in \mathcal{F}$:

$$\left| \frac{\sum_{x \in S} \omega(x) f(x)}{\sum_{x \in X} \mu(x) f(x)} - 1 \right| \le \varepsilon. \tag{5}$$

Sampling the subset $S$ using a $k$-DPP with marginal kernel $K \in \mathbb{R}^{n \times n}$ provides significant advantages over random sampling. In practice, we construct the kernel $K$ using the cosine similarity between ViT embeddings, ensuring computational efficiency while maintaining diversity. The key benefit lies in the variance structure: DPPs naturally select diverse points, leading to representative estimates.

---

**Algorithm 3** Physics-Based CCE Repulsion Dynamics

1: **Input:** Previous embeddings $\{\mathcal{P}_1, \dots, \mathcal{P}_{\ell-1}\}$, repulsive constant $\zeta$, mass $m$, time step $\Delta t$, duration $\tau$
2: **Output:** Current task embeddings $\mathcal{P}_\ell$
3: Initialize $\mathcal{P}_\ell^0 \leftarrow \phi(\boldsymbol{w}_{\ell-1}, D_\ell)$, velocities $\boldsymbol{v}_\ell^j \leftarrow \mathbf{0}$
4: **for** $t = 1$ to $\tau$ **do**
5:     **for** $j = 1$ to $J_\ell$ **do**      ▷ Current task classes
6:         $\boldsymbol{F}_\ell^j \leftarrow \mathbf{0}$      ▷ Reset force accumulator
7:         **for** $(i', j') \in \{(i', j') \mid i' \le \ell, \, j' \le J_{i'}\} \setminus \{(\ell, j)\}$ **do**
8:             $\boldsymbol{d} \leftarrow \boldsymbol{p}_\ell^j - \boldsymbol{p}_{i'}^{j'}$
9:             $\boldsymbol{F}_\ell^j \leftarrow \boldsymbol{F}_\ell^j + \dfrac{\zeta}{\|\boldsymbol{d}\|^3} \boldsymbol{d}$      ▷ Repulsive force
10:         **end for**
11:         $\boldsymbol{v}_\ell^j \leftarrow \boldsymbol{v}_\ell^j + \frac{\boldsymbol{F}_\ell^j}{m} \Delta t$      ▷ Update velocity
12:         $\boldsymbol{p}_\ell^j \leftarrow \boldsymbol{p}_\ell^j + \boldsymbol{v}_\ell^j \Delta t$      ▷ Update position
13:     **end for**
14: **end for**

---

For any bounded test function $\varphi : X \to \mathbb{R}$, define $\Lambda(\varphi) = \sum_{x \in S} \varphi(x)$ as the empirical sum over the selected subset. The variance of this estimator is:

$$\mathrm{Var}\big[\Lambda(\varphi)\big] = \mathrm{Tr}\big[\Phi(I - K)\Phi K\big], \tag{6}$$

where $\Phi$ is the diagonal operator defined as $(\Phi g)(x) = \varphi(x)g(x)$. Crucially, the factor $(I - K)$ explicitly encodes *negative dependence*–when one point is selected, similar points become less likely to be chosen. This significantly reduces variance compared to independent (i.i.d.) sampling, ensuring that our small coreset provides more reliable estimates of the full dataset statistics.

This variance reduction translates into strong concentration guarantees. For a universal constant $A > 0$, the DPP sampler satisfies:

$$\Pr\Big( |\Lambda(\varphi) - \mathbb{E}\Lambda(\varphi)| \geq \varepsilon \Big) \;\leq\; 2\exp\left( -\frac{\varepsilon^2}{4A\operatorname{Var}\Lambda(\varphi)} \right). \tag{7}$$

This provides Hoeffding-type concentration bounds without requiring independence assumptions, ensuring diverse replay with minimal storage requirements (see Algorithm 2).

The third pillar of ReMoBA inspired by Nori et al. (2023); van de Ven et al. (2021) further mitigates inter-client task confusion via a mixture of batchnoise autoencoders (MBAs), in which each artificial neuron is characterized as $f\left(\sum_i w_i x_i + b\right) + \alpha\tilde{z}$, where $\tilde{z}$ introduces regularizing noise with magnitude controlled by hyperparameter $\alpha$ and normalized noise term $\tilde{z} \sim \mathcal{N}(0,1)$ drawn from a standard Gaussian distribution.

Following Nori et al. (2023); van de Ven et al. (2021), instead of a single autoencoder, ReMoBA utilizes a collection of lightweight, class-conditional batchnoise autoencoders. Crucially, these autoencoders operate *not* on raw input data, but on the rich, high-dimensional feature representations extracted by the shared ViT PTM backbone, $\phi(\cdot)$. This design makes them highly efficient to train and allows them to focus on modeling the specific data manifold of each class within a semantically meaningful space.

For each class $c \in \mathcal{Y}$ encountered by client $k$ in task $t$, a dedicated autoencoder, consisting of a batchnoise encoder $E_c^z$ and a batchnoise decoder $G_c^z$, is trained. The objective is to reconstruct the feature vector of a given sample. The reconstruction loss $\mathcal{L}_{\text{recon}}^{(c)}$ is as follows:

$$\mathbb{E}_{\boldsymbol{x} \in D_{k,c}^t} \| \phi(\boldsymbol{x}) - G_c^z(E_c^z(\phi(\boldsymbol{x}))) \|_2^2, \tag{8}$$

where $D_{k,c}^t$ is the subset of data from client $k$'s task $t$ belonging to class $c$.

Using activation noise during the inference phase, MBAs transform a deterministic prediction into a stochastic sampling process. The theoretical goal is to compute the expected log-likelihood $l_{\boldsymbol{\theta}^*}(\boldsymbol{x}|y)$ over a continuous noise distribution $r(z)$, as defined by:

$$\mathbb{E}_{z \sim r(z)}[\log p_{\boldsymbol{\theta}^*}(\boldsymbol{x}|y,z)] = \int_z \log p_{\boldsymbol{\theta}^*}(\boldsymbol{x}|y,z)r(z)dz. \tag{9}$$

---

**Algorithm 4** ReMoBA Server Protocol

1: **Input:** $K$ tasks, $I$ clients per round, $\Omega$ communication rounds
2: **Initialize:** PTM $\phi_0$, $\mathcal{P} \leftarrow \emptyset$
3: **for** task $h = 1$ to $K$ **do**
4:     Select $I$ clients $\mathcal{C}_h$; broadcast $\phi_{h-1}$
5:     Receive embeddings $\{\mathcal{P}_h^c\}_{c \in \mathcal{C}_h}$ from clients
6:     $\mathcal{P}_h^{aligned} \leftarrow$ PARTICLE_REPULSION($\{\mathcal{P}_h^c\}$)
7:     Broadcast $\mathcal{P}_h^{aligned}$ to $\mathcal{C}_h$
8:     **for** round $\omega = 1$ to $\Omega$ **do**
9:         Receive $\{\phi_h^c\}_{c \in \mathcal{C}_h}$ from clients
10:         Aggregate $\phi_h^{global} \leftarrow \frac{1}{I}\sum_{c \in \mathcal{C}_h} \phi_h^c$
11:         Broadcast $\phi_h^{global}$ to $\mathcal{C}_h$
12:     **end for**
13: **end for**

---

**Algorithm 5** ReMoBA Client Update

1: **Input:** PTM $\phi_{h-1}^{global}$, local data $D_h^c$, coresets $\{\mathcal{S}_i^c\}_{i < h}$
2: **Output:** Model update $\phi_h^c$, $\{\mathcal{A}_j^c\}$
3: $\mathcal{P}_h^c \leftarrow$ EXTRACT_EMBEDDINGS($\phi_{h-1}^{global}, D_h^c$)   ▷ send to server
4: Receive $\mathcal{P}_h^{aligned}$ from server
5: $\mathcal{S}_h^c \leftarrow$ DPP_CORESET_SAMPLE($D_h^c$)
6: $\mathcal{M}_h^c \leftarrow \bigcup_{i \leq h} \mathcal{S}_i^c$
7: $\phi_h^c \leftarrow \phi_{h-1}^{global}$
8: **for** round $r = 1$ to $R$ **do**
9:     $\mathcal{L} \leftarrow \lambda_1 \big\| \phi_h^c(\mathcal{M}_h^c) - \{\mathcal{P}_h^c\} \big\|^2$
            $+ \lambda_2 \big\| \phi_h^c(D_h^c) - \mathcal{P}_h^{aligned} \big\|^2$
10:     $\phi_h^c \leftarrow \phi_h^c - \eta\nabla\mathcal{L}$
11: **end for**
12: $\{\mathcal{A}_j^c\} \leftarrow$ TRAIN_MIXTUREAUTOENCODERS($D_h^c$)
13: **return** $\phi_h^c$, $\{\mathcal{A}_j^c\}$

---

However, since this integral is intractable, Nori et al. (2023); van de Ven et al. (2021) approximate it using a Monte Carlo estimation. For a given input $\boldsymbol{x}$, the model's response is evaluated $n$ times, with each forward pass perturbed by an independent noise vector $z^{(i)}$ drawn from the inference noise distribution. The log-likelihoods from these $n$ stochastic evaluations are then averaged to produce a single estimate:

$$l_{\boldsymbol{\theta}^*}(\boldsymbol{x}|y) \approx \frac{1}{n}\sum_{i=1}^n \log p_{\boldsymbol{\theta}^*}(\boldsymbol{x}|y,z^{(i)}). \tag{10}$$

Classification in MBAs is done by a direct application of Bayes' rule, where the model assigns the input $\boldsymbol{x}$ to the class $\hat{y}$ that yields the maximum estimated log-likelihood value: $\hat{y} = \operatorname*{argmax}_y l_{\boldsymbol{\theta}^*}(\boldsymbol{x}|y)$.

Table 1: Average ($\bar{\mathcal{A}}$) and final ($\mathcal{A}_B$) accuracy (%) of all methods on federated DIL benchmarks (first task order). Best results are **bold**. All methods use ViT-B/16 IN1K. Methods with † use exemplars (10/class). Office-Home and DomainNet report for 10 and 20 clients; CORe50 and CDDB-Hard report 10-client results.

| Methods/Benchmarks | Office-Home | | | | DomainNet | | | | CORe50 | | CDDB-Hard | |
|---|---|---|---|---|---|---|---|---|---|---|---|---|
| Clients | 10 | | 20 | | 10 | | 20 | | 10 | | 10 | |
| Accuracy | $\bar{\mathcal{A}}$ | $\mathcal{A}_B$ | $\bar{\mathcal{A}}$ | $\mathcal{A}_B$ | $\bar{\mathcal{A}}$ | $\mathcal{A}_B$ | $\bar{\mathcal{A}}$ | $\mathcal{A}_B$ | $\bar{\mathcal{A}}$ | $\mathcal{A}_B$ | $\bar{\mathcal{A}}$ | $\mathcal{A}_B$ |
| FedAvg (McMahan et al., 2017) | 68.63 | 70.71 | 64.99 | 68.05 | 43.14 | 44.28 | 39.50 | 41.62 | 69.59 | 71.66 | 45.32 | 45.75 |
| FedReplay† (Pennisi et al., 2024) | 76.00 | 79.27 | 73.26 | 75.90 | 60.06 | 61.35 | 57.32 | 58.98 | 79.73 | 81.01 | 45.07 | 45.16 |
| iCaRL† (Rebuffi et al., 2017) | 71.65 | 76.38 | 70.09 | 74.32 | 54.27 | 55.40 | 52.71 | 53.82 | 71.78 | 76.17 | 48.23 | 48.44 |
| MEMO† (Zhou et al., 2022) | 65.23 | 66.64 | 61.85 | 62.07 | 55.71 | 57.18 | 52.33 | 52.60 | 59.92 | 63.11 | 49.96 | 58.38 |
| SimpleCIL (Zhou et al., 2023) | 66.59 | 70.29 | 63.44 | 67.56 | 34.60 | 38.65 | 31.45 | 35.92 | 65.99 | 69.32 | 54.52 | 62.81 |
| L2P (Wang et al., 2022c) | 69.93 | 73.83 | 66.54 | 69.68 | 44.44 | 46.46 | 41.05 | 42.31 | 78.38 | 82.99 | 58.69 | 67.89 |
| DualPrompt (Wang et al., 2022b) | 70.91 | 76.28 | 68.48 | 72.38 | 46.95 | 48.29 | 44.39 | 44.52 | 80.21 | 82.91 | 54.37 | 62.81 |
| CODA-Prompt (Smith et al., 2023) | 76.43 | 79.74 | 73.70 | 77.24 | 53.72 | 55.78 | 50.99 | 51.28 | 82.83 | 86.07 | 58.40 | 66.92 |
| EASE (Zhou et al., 2024c) | 73.74 | 71.67 | 71.33 | 72.82 | 48.22 | 50.15 | 45.81 | 48.32 | 81.92 | 83.57 | 59.46 | 67.08 |
| RanPAC (McDonnell et al., 2023) | 74.16 | 77.26 | 70.56 | 74.45 | 48.92 | 49.95 | 45.32 | 47.14 | 74.28 | 77.36 | 61.78 | 67.74 |
| S-iPrompt (Wang et al., 2022a) | 72.23 | 75.08 | 68.22 | 72.83 | 53.12 | 53.54 | 49.11 | 51.29 | 80.91 | 82.09 | 67.06 | 76.37 |
| DUCT (Zhou et al., 2024b) | 80.31 | 80.34 | 77.47 | 77.98 | 60.10 | 62.25 | 57.76 | 57.90 | 84.40 | 86.76 | 70.22 | 75.32 |
| **ReMoBA (Ours)** | **80.55** | **81.61** | **78.14** | **78.72** | **62.07** | **63.34** | **59.66** | **60.45** | **87.31** | **89.86** | **73.46** | **80.02** |
| ReMoBA-Cos (w/ Cos Classifier) | 78.21 | 79.03 | 75.88 | 76.41 | 59.84 | 60.92 | 57.11 | 58.03 | 85.12 | 87.34 | 70.89 | 76.45 |
| ReMoBA-Rand (Random Sampling) | 77.64 | 78.29 | 75.12 | 75.93 | 58.73 | 59.87 | 56.05 | 57.21 | 83.97 | 85.88 | 69.55 | 74.88 |
| ReMoBA-Herd (Herding Sampling) | 79.02 | 80.15 | 76.73 | 77.34 | 60.45 | 61.76 | 58.22 | 59.10 | 85.83 | 88.02 | 71.67 | 77.23 |
| ReMoBA-LJ (Lennard-Jones) | 79.88 | 80.94 | 77.51 | 78.10 | 61.22 | 62.41 | 58.97 | 59.83 | 86.74 | 88.95 | 72.31 | 78.67 |

It is worth noting that as mentioned in the previous section, following prior work for each domain-class combination, one batchnoise autoencoder is trained. Algorithms 4 and 5 outline the procedures of ReMoBA at the server and clients.

## 5 EXPERIMENTS

**Experimental Setting:** We evaluate ReMoBA in the Federated Domain-Incremental Learning (FDIL) setting, following established benchmarks and protocols from both PTM-based domain-incremental learning (Wang et al., 2022b; Zhou et al., 2024b; Smith et al., 2023; Zhou et al., 2023) and federated continual learning literature (Li et al., 2019; Dong et al., 2022; Nori et al., 2025). Experiments are conducted on four widely-used benchmarks: **Office-Home** (Venkateswara et al., 2017), **DomainNet** (Peng et al., 2019), **CORe50** (Lomonaco & Maltoni, 2017), and **CDDB-Hard** (Li et al., 2023), covering diverse visual domains, real-world object categories, and challenging distribution shifts. Each dataset is partitioned across 10 (and 20) clients using non-IID splits, where tasks correspond to the sequential arrival of new domains per client, simulating realistic federated deployment. All methods are evaluated under five distinct task orders to mitigate order bias. Full experimental details, including client-domain assignment, hyperparameters, memory budgets, and additional splits, are provided in the supplementary material.

**Baselines and Evaluation Metrics:** We compare ReMoBA against state-of-the-art methods from both federated continual learning (FedAvg (McMahan et al., 2017), FedReplay (Pennisi et al., 2024), MOON (Li et al., 2021), FedWeIT (Yoon et al., 2021)) and centralized domain-incremental learning (Replay (Ratcliff, 1990), iCaRL (Rebuffi et al., 2017), MEMO (Zhou et al., 2022), SimpleCIL (Zhou et al., 2023), L2P (Wang et al., 2022c), DualPrompt (Wang et al., 2022b), CODA-Prompt (Smith et al., 2023), EASE (Zhou et al., 2024c), RanPAC (McDonnell et al., 2023), S-iPrompt (Wang et al., 2022a), DUCT (Zhou et al., 2024b)). All methods use the same ViT-B/16 backbone pre-trained on ImageNet-21K to ensure fair comparison. Performance is measured by two key metrics: (1) **Average Accuracy** ($\bar{\mathcal{A}}$), mean accuracy across all incremental tasks; and (2) **Final Accuracy** ($\mathcal{A}_B$), performance after learning the last task.

**Implementation Details:** We use SGD with momentum 0.9, batch size 128, and initial learning rate 0.001 (decayed by 0.1 every 5 epochs), training for 15 local epochs per task. Server aggregation occurs every 3 local epochs (10 rounds per task). Exemplar-based methods, including ReMoBA, use a fixed memory budget of 10 samples per class. Hyperparameters for DPP sampling ($\varepsilon = 0.1$), repulsion dynamics ($\zeta = 1.0, \tau = 50, \Delta t = 0.01$), and MBA noise ($\alpha = 0.9, n = 10$ Monte Carlo samples) are kept consistent across datasets.

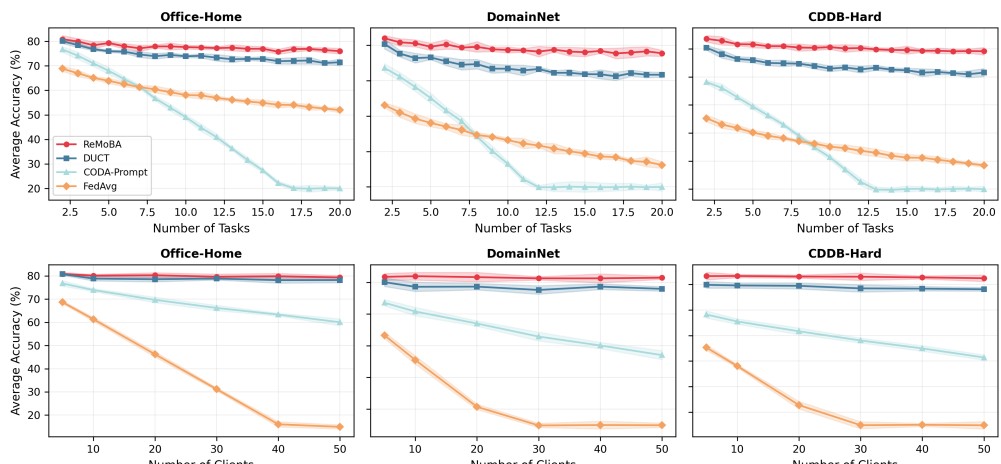

Figure 1: **Scalability of ReMoBA under increasing task and client load. (Top row)** Average accuracy as the number of tasks increases from 1 to 20 (fixed 10 clients). **(Bottom row)** Performance as the number of clients scales from 5 to 50 (fixed 5 tasks). ReMoBA maintains high accuracy across both axes, demonstrating strong resistance to catastrophic forgetting and inter-client misalignment.

**Main Results:** As shown in Table 1, ReMoBA consistently achieves state-of-the-art performance across all benchmarks and client scales. On Office-Home (10 clients), ReMoBA improves average accuracy by 0.24% and final accuracy by 1.27% over the strongest baseline, DUCT. Gains are more pronounced on larger-scale and more heterogeneous benchmarks: on DomainNet (10 clients), ReMoBA achieves 1.97% higher $\bar{\mathcal{A}}$ and 1.09% higher $\mathcal{A}_B$; on CDDB-Hard, improvements reach 3.24% in $\bar{\mathcal{A}}$ and a remarkable 4.70% in $\mathcal{A}_B$. These results validate ReMoBA's ability to mitigate both intra-client forgetting (via DPP replay) and inter-client confusion (via physics-based alignment and MBAs), particularly under severe domain shifts and client heterogeneity.

Notably, pure federated methods (e.g., FedAvg) suffer from significant performance degradation due to misaligned representations and lack of replay, while centralized DIL methods (e.g., iCaRL, L2P) struggle to generalize across non-IID client distributions. Even recent PTM-based approaches like DUCT and CODA-Prompt, which excel in centralized settings, underperform in FDIL due to their inability to explicitly model inter-client task confusion or adapt representations globally.

**Ablation Analysis:** To dissect the contribution of each ReMoBA component, we evaluate four key ablations reported in Table 1. First, *ReMoBA-Cos*, which replaces the Mixture of BatchNoise Autoencoders (MBAs) with a standard cosine classifier, shows a consistent performance drop across all datasets, for instance, a *2.3%* decline in average accuracy on CDDB-Hard, confirming that stochastic MBAs are better for resolving inter-client task confusion and improving robustness under domain shift. Second, *ReMoBA-Rand*, which substitutes DPP-based exemplar selection with random sampling, exhibits degraded performance (e.g., *2.9% $\bar{\mathcal{A}}$ on CORe50*), demonstrating that diversity-aware selection is more effective for preserving previous knowledge and combating intra-client forgetting. Third, *ReMoBA-Herd*, which adopts iCaRL-style herding (centroid-based selection), performs better than random sampling but still underperforms DPP (e.g., *1.5% $\bar{\mathcal{A}}$ on DomainNet-10*), as herding prioritizes centroid proximity over diversity, leading to redundancy in the latent space. Finally, *ReMoBA-LJ*, which replaces Coulomb repulsion with the Lennard-Jones potential (introducing both attractive and repulsive forces), results in a slight but consistent performance degradation (e.g., *0.7% $\bar{\mathcal{A}}$ on Office-Home-10*).

**Scalability Analysis:** To evaluate the scalability of ReMoBA in realistic FDIL settings, we analyze its performance under two key stress tests: increasing task sequence length and growing client population. As shown in Figure 1, ReMoBA exhibits remarkable robustness in both scenarios. In the top row, when the number of tasks increases from 1 to 20, ReMoBA's average accuracy declines by only ~2% across all benchmarks, indicating effective mitigation of catastrophic forgetting. In contrast, CODA-Prompt and FedAvg show significant degradation—especially on CDDB-Hard,

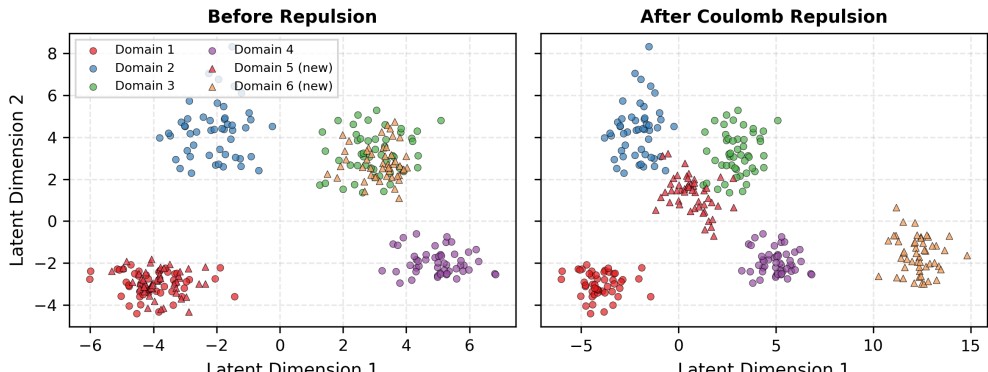

Figure 2: **Latent space embedding dynamics under Coulomb repulsion.** This visualization illustrates the effect of ReMoBA's physics-inspired alignment mechanism on domain embeddings in the latent space of a pre-trained ViT backbone. **(Left)** Before applying repulsive forces, the two newly introduced domains (e.g., Sketch and Infograph, marked with triangles) are embedded close to existing domain clusters, leading to potential inter-client task confusion and representation overlap. **(Right)** After the server applies Coulomb repulsion dynamics — where each new domain embedding experiences a force proportional to the inverse square of its distance from all prior domains — the new domains are pushed away into distinct regions while preserving intra-domain compactness. The data shown is synthetic but representative of real-world FDIL scenarios, generated using a non-IID split of the Office-Home benchmark across 3 clients, with each client receiving sequential domain tasks. Embeddings are extracted from the last layer of a ViT-B/16 model pre-trained on ImageNet-21K.

where CODA-Prompt drops by over 15%—highlighting their vulnerability to sequential domain shifts.

The bottom row reveals ReMoBA's resilience to client heterogeneity. Even with 50 clients, ReMoBA maintains accuracy within 1–2% of its baseline, thanks to global repulsive force alignment that enforces consistent representations across clients. FedAvg, however, collapses to near-random performance (below 20%) as client diversity increases, due to misaligned local updates and lack of coordination. DUCT performs better than FedAvg but still degrades significantly, underscoring the importance of explicit inter-client consistency mechanisms. These results confirm that ReMoBA is not only effective in small-scale settings but also scalable to large, dynamic federated environments.

**Qualitative Representation Analysis:** Figure 2 provides qualitative validation of ReMoBA's alignment mechanism. The clear separation of new domains after repulsion visually confirms the method's ability to resolve inter-client confusion—a core challenge in FDIL that competing methods fail to address explicitly. This structural regularization in the latent space complements the quantitative gains observed in Table 1 and Figure 1, offering interpretable evidence of representation advantage.

## 6 CONCLUSION

In this work, we introduce ReMoBA, a theoretically grounded and empirically effective approach for Federated Domain-Incremental Learning with pre-trained models. ReMoBA addresses the critical challenges of inter-client and intra-client task confusion as well as representation misalignment through three synergistic components: (1) diversity-guaranteed DPP-based replay, (2) physics-inspired Coulomb repulsion for global embedding alignment, and (3) mixture of batchnoise autoencoders for robust, generative classification. Extensive experiments across four benchmarks demonstrate consistent state-of-the-art performance, with significant gains in both average and final accuracy—particularly under large-scale, non-IID, and long-task-sequence conditions. Ablation studies and visualizations confirm the necessity of each component and provide interpretable insights into the method's behavior. ReMoBA offers a scalable, practical, and principled solution for deploying PTMs in real-world federated continual learning systems.

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
