# OpenReview forum: "ReMoBA: Representative Replay and Mixture of BatchNoise Autoencoders for Pre-Trained Model-Based Federated Domain-Incremental Learning"
_ICLR.cc/2026/Conference — ICLR 2026 Conference Withdrawn Submission_

### Official Review · Reviewer_tdb7 · 2025-10-26

**Soundness:** 2
**Presentation:** 2
**Contribution:** 2
**Rating:** 4
**Confidence:** 4

**Summary:**

This paper introduces ReMoBA (Representative Replay and Mixture of BatchNoise Autoencoders), a framework designed for Federated Domain-Incremental Learning (FDIL) using pre-trained models (PTMs). FDIL is a challenging scenario where multiple clients sequentially learn domain-specific tasks. The paper identifies three major problems — inter-client task confusion, intra-client task confusion, and catastrophic forgetting — and proposes ReMoBA to address them through three components.

First, a Determinantal Point Process (DPP)-based coreset sampler ensures diversity in replayed exemplars, effectively preserving prior domain knowledge. Second, a Coulomb repulsion–based embedding alignment mechanism at the server aligns representations across clients by modeling global domain embeddings as charged particles repelling each other to reduce representational overlap. Third, a Mixture of BatchNoise Autoencoders (MBAs) introduces structured activation noise for class-conditional generative replay, improving robustness against inter-client confusion and domain shifts.

Empirical evaluations on four vision benchmarks (Office-Home, DomainNet, CORe50, CDDB-Hard) using ViT-B/16 demonstrate that ReMoBA consistently outperforms prior FDIL and centralized domain-incremental baselines. However, the paper’s weaknesses limit its overall impact. All experiments are restricted to vision-only ViT models, leaving the applicability of ReMoBA to other modalities (e.g., NLP, speech, multimodal PTMs) untested. The experimental gains are modest (often 1–2%), and no computational overhead analysis is provided, which weakens empirical credibility. The method’s algorithmic complexity—combining DPP sampling, repulsive-force optimization, and multiple generative autoencoders—may also challenge practical deployment in federated settings. Moreover, the work overrelies on ViT pre-training, and the improvements might partly arise from the backbone rather than the proposed mechanisms. These limitations make the contributions somewhat narrow and experimentally underexplored, despite the strong theoretical motivation and sound design.

**Strengths:**

Good combination of ideas: The integration of DPP-based exemplar selection, physics-inspired repulsive alignment, and stochastic generative replay (MBAs) forms an elegant hybrid of mathematical and biologically inspired techniques.

Good theoretical grounding: The paper formalizes inter/intra-client confusion mathematically and connects them to the stability–plasticity dilemma in continual learning.

**Weaknesses:**

1.	Limited domain scope: All experiments focus on vision tasks with ViT backbones, limiting the claimed generality for broader FDIL (e.g., NLP or multimodal PTMs). The effectiveness for non-visual modalities remains uncertain.

2.	Experimental depth: While quantitative results are comprehensive, qualitative and statistical significance analyses (e.g., variance over seeds) are missing. The improvement margins are sometimes modest (1–2%), which weakens the empirical persuasiveness.

3.	Complexity vs. practicality: The combination of DPP sampling, Coulomb repulsion, and MBAs increases algorithmic and communication complexity, but the paper lacks runtime or communication cost comparisons.

4.	Overreliance on PTMs: Although it claims theoretical novelty, much of the performance gain may stem from the use of large ViT encoders, not necessarily from ReMoBA itself.

5.	Missing broader discussion: The ethical, data privacy, or convergence implications of server-side alignment (with repulsive dynamics) are not deeply analyzed, which could be relevant for federated systems.

**Questions:**

See Weakness.

---

### Official Review · Reviewer_Gb8r · 2025-10-27

**Soundness:** 2
**Presentation:** 2
**Contribution:** 1
**Rating:** 2
**Confidence:** 5

**Summary:**

This paper proposes ReMoBA, a method combining Representative Replay and Mixture of BatchNoise Autoencoders for pre-trained model-based Federated Domain-Incremental Learning (FDIL). It introduces a DPP-based exemplar selection strategy to ensure diversity, a physics-inspired Coulomb repulsion mechanism for global domain alignment, and class-wise batch-noise autoencoders to reduce inter-client confusion. Experimental results show improved accuracy and robustness over existing FDIL and continual learning methods.

**Strengths:**

- The paper presents a well-organized combination of replay and generative techniques tailored for pre-trained models in FDIL.
- Extensive experiments across multiple datasets demonstrate consistent performance improvements and scalability.

**Weaknesses:**

- The claimed “theoretical grounding” of the approach is not rigorously supported. The mathematical formulation is descriptive rather than provably analytical.
- The method’s complexity (DPP selection + Coulomb repulsion + mixture autoencoders) makes it difficult to reproduce and may not scale well in practical federated settings.
- The novelty is limited, as each major component (DPP replay, repulsive alignment, generative autoencoders) has been used in similar contexts before.
- The improvements in accuracy are relatively small and may not justify the added algorithmic and computational complexity.

**Questions:**

- How does ReMoBA compare to simpler exemplar-based replay methods in terms of computation and communication overhead?
- What is the specific benefit of using Coulomb repulsion instead of other embedding regularization or metric-learning techniques?
- How sensitive are the results to the number of exemplars and the DPP hyperparameters?
- How is privacy preserved when transmitting embeddings and autoencoder outputs between clients and the server?
- What are the computational costs and memory requirements per client compared to other FDIL baselines?
- Is the repulsive force algorithm stable when the number of domains or clients grows significantly?

---

### Official Review · Reviewer_Gxfe · 2025-11-01

**Soundness:** 2
**Presentation:** 1
**Contribution:** 1
**Rating:** 2
**Confidence:** 5

**Summary:**

The paper proposes ReMoBA, an approach that guarantees sample diversity by leveraging a latent-space exemplar selection strategy to replay a tiny curated subset of past data stored on the client side. The authors identify inter-client, intra-client, and within-task confusion as the main challenges when using pre-trained models (PTMs) in Federated Domain-incremental Learning (FDIL). Experiments on Office-Home, DomainNet, CORe50, and CDDB-Hard benchmarks show that ReMoBA outperforms state-of-the-art methods.

**Strengths:**

- The problem studied in this work is timely and important.
- The structure of the paper is easy to follow.
- Consistent state-of-the-art results across multiple FDIL benchmarks and scalability analyses.

**Weaknesses:**

- Overall, the paper’s contribution is limited, with verbose writing, inconsistent notations, excessive definitions and assumptions, and weakly motivated research questions.
- The representation and structure require significant improvement, and the work lacks both theoretical and experimental analyses to substantiate its novelty.
- The discussion of inter- and intra-client task confusion is insufficiently supported, relying mainly on centralized CL references rather than federated settings. Moreover, the concept of task confusion overlaps with task-free FCL, yet the authors fail to review or cite recent works in task-free CL and FCL.
- SOTA works in coreset selection for CL is missing.
- Algorithms 2 and 3 are unclear, poorly connected, and lack theoretical justification. For instance, cosine similarity in Algorithm 2 involves concatenated features, which implies a high computational cost, and the parameter α in line 3 is not clearly defined.
- The discussion of batch noise autoencoders is also incomplete; their training and role within the system are not elaborated, and they do not constitute a novel contribution.
- The framework involves several heavy components (DPP sampling, repulsion dynamics, multiple autoencoders), but the analysis of computational and communication overhead is not discussed.

**Questions:**

- Why are incremental strategies divided into discriminative and generative? What do the authors mean by discriminative and generative in these scenarios?
- Why are the loss functions in Eqs. (1), (2), (3), and (4) are totally different? What are the rationales behind?
- How can we characterize the "cross-client" loss $I_\theta$ as in Eq. (3)?
- Are the $k$ in Eq. (3) (4) the client index as mentioned in Section 3? Then, why are the indices $k,m,n$?
- How is the eigendecomposition in Algo 2 / line 5 computed?
- What is the second pillar of ReMoBA?
- How is Eq. (8) placed in the FCL system? It does not appear anywhere in the algorithms.
- How are the baselines of centralized CL set up in the experimental evaluations?
- Please explain why the results of centralized CL are lower than those of FCL?
- Why are the results in Figure 1 almost linear?

---

### Official Review · Reviewer_KPhY · 2025-11-06

**Soundness:** 3
**Presentation:** 2
**Contribution:** 2
**Rating:** 2
**Confidence:** 5

**Summary:**

The paper targets Federated Domain-Incremental Learning (FDIL) with pre-trained models, focusing on inter-/intra-client task confusion and catastrophic forgetting. It proposes ReMoBA, combining: 1) A DPP-based latent-space exemplar selection; 2) A physics-inspired Coulomb repulsion scheme to align domain embeddings across clients; 3) A mixture of batch-noise autoencoders (MBAs) as a generative classifier on PTM features. Experiments on Office-Home, DomainNet, CORe50, and CDDB-Hard with ViT-B/16 backbones show improvements over several PTM-based continual/federated baselines.

**Strengths:**

1. The paper targets federated domain-incremental learning with strong reliance on pretrained models, focusing on task confusion and forgetting in realistic cross-client, cross-domain settings. This is an important and under-explored problem with clear practical relevance.

2. ReMoBA integrates three complementary components, diverse exemplar selection via DPP-based coresets, representation alignment/repulsion in the embedding space, and a generative MBA-based classifier, forming a conceptually coherent pipeline of “sample selection + feature space regularization + decision refinement.”

3. The method is evaluated on multiple standard benchmarks and against a range of strong continual/federated baselines under a unified pretrained ViT-B/16 setting.

**Weaknesses:**

1. The paper strongly emphasizes three types of confusion (inter-client, intra-client, within-task) and catastrophic forgetting, but the mapping from each ReMoBA component to each phenomenon is mostly narrative, not rigorously established.

2. Eqs. (3)–(4) are presented as a “mathematical characterization” of FDIL/task confusion but notations (M, N, k, m, n, l, etc.) are never clearly defined. The connection between these sums and actual FDIL training dynamics is unclear. The decomposition into “local” vs “global” forgetting is asserted rather than derived. As a result, the “theoretical insight” claim feels overstated. This is a key weakness for a theory-flavored paper.

3. Each major component of ReMoBA is adapted from existing ideas, DPP-based coresets for diverse exemplar selection, physics-inspired repulsion for embedding separation, and batch-noise autoencoder-based generative classification. The main novelty therefore lies in combining these modules within an FDIL setting. However, the paper does not sufficiently analyze how these components interact or why their joint use is fundamentally necessary, so the method risks being perceived as a stacked collection of techniques rather than a unified framework driven by a clear core principle.

4. The ε-multiplicative coreset condition (Eq. 5) is cited from Bardenet et al., but the paper does not specify the underlying function class, weighting scheme, or how this theory aligns with the actual FDIL loss. The required assumptions (e.g., boundedness, PSD kernel) are not verified, and the variance/concentration results in Eqs. (6)–(7) are stated without proof or careful adaptation to the proposed cosine-kernel, scaled, Nyström-based sampler. As written, the method is presented as if it enjoys formal guarantees that are not rigorously justified, making the theoretical claims appear overstated.

5. Eq. (9)–(10) is standard Monte Carlo estimation, but the classification rule’s integration into the federated protocol is underspecified.

6. Notation is inconsistent and sometimes confusing. For example, K is used for both the number of clients and tasks, T and B appear with different meanings (tasks, blocks, etc.), and I is overloaded as an indicator and possibly identity. In addition, the same symbol P is used for both domain-level and class-level embeddings, which makes Algorithms 3–5 difficult to follow. A concise notation table and stricter symbol discipline are needed to improve clarity and reproducibility.

7. Most of the compared baselines are originally designed for centralized continual/domain-incremental learning rather than for federated settings, which weakens the fairness and relevance of the evaluation. The authors are encouraged to include recent methods explicitly tailored for federated continual/domain-incremental learning as baselines, or to provide a clear justification and careful adaptation protocol when repurposing centralized methods to FL scenarios.

8. Algorithm 3 introduces a Coulomb-style repulsion on embeddings, but it is unclear whether these represent class prototypes, domain embeddings, or something else, as the text switches terms inconsistently. The procedure does not discuss normalization or safeguards against numerical instability, nor does it provide any analysis of convergence or invariance. The Lennard-Jones variant is only presented empirically without explaining why one potential is preferable to the other, so the repulsion mechanism overall comes across as heuristic rather than theoretically grounded.

**Questions:**

1. How do the cited DPP coreset and concentration results rigorously apply to the practical sampling pipeline used in ReMoBA, given the specific kernel choices and approximations?

2. Can the authors provide a more systematic analysis to show why combining DPP sampling, repulsion, and MBA is fundamentally necessary, beyond incremental gains from each component alone?

3. Why are there no comparisons with recent federated continual/domain-incremental learning methods, and can the authors report communication and computation overheads to justify the practicality of ReMoBA?

4. Could the authors more precisely define the embeddings used in the repulsion module, the construction of aligned prototypes, and the exact loss formulations, so that the training pipeline can be faithfully reproduced?


If the authors can substantially clarify the theoretical claims, tighten the notation and method description for full reproducibility, include stronger federated-specific baselines with overhead analysis, and better justify the interaction and necessity of each module, I would be inclined to reconsider my rating upward.

---

### Note · Authors · 2025-11-13

**Comment:**

We withdraw this submission due to low scores. We thank the reviewers for their time and constructive comments.

**Withdrawal Confirmation:**

I have read and agree with the venue's withdrawal policy on behalf of myself and my co-authors.